# Hypoadrenocorticism in a Dog Following Recovery from Alpha-Amanitin Intoxication

**DOI:** 10.3390/vetsci10080500

**Published:** 2023-08-03

**Authors:** Emily A. Cohen, Courtney M. Moeller, Jonathan D. Dear

**Affiliations:** 1William Pritchard Veterinary Medical Teaching Hospital, School of Veterinary Medicine, University of California Davis, One Shields Avenue, Davis, CA 95616, USA; emcohen@ucdavis.edu (E.A.C.); cmwrig@ucdavis.edu (C.M.M.); 2Department of Medicine and Epidemiology, School of Veterinary Medicine, University of California Davis, One Shields Avenue, Davis, CA 95616, USA

**Keywords:** amatoxin, alpha-amanitin, adrenal insufficiency, hepatic dysfunction, hypoadrenocorticism, hepatoadrenal syndrome

## Abstract

**Simple Summary:**

Amanitin-containing mushroom poisonings occur worldwide and affect both humans and animals. Intoxication from these mushrooms can cause severe gastrointestinal, liver, and neurologic dysfunction, which can result in death. This case report discusses amanitin-induced liver dysfunction and the development of hypoadrenocorticism following recovery in a dog. A clearer understanding of post-intoxication outcomes would help veterinarians prepare pet owners to understand the implications and long-term consequences of amanitin intoxication.

**Abstract:**

A 10-year-old, female spayed Labrador Retriever was referred for acute hepatopathy and urinary retention. Blood work from the initial presentation (day 0) revealed a severe, mixed hepatopathy. Over the course of the patient’s hospitalization, the patient developed liver insufficiency. Urine was submitted for toxicological screening and revealed detection of a trace concentration of alpha-amanitin. The patient was treated supportively for alpha-amanitin intoxication and was discharged from the hospital on day 8, with most biochemical parameters being markedly improved. The patient was persistently hyporexic at the time of discharge. On day 15, at a recheck appointment, the patient had lost 2.4 kg and liver enzymology revealed improved values. On day 24, the patient was presented for anorexia and vomiting and had lost another 2.3 kg. Blood work and endocrinological testing at that time were consistent with hypoadrenocorticism. The patient was started on glucocorticoids and mineralocorticoids. At day 106, the patient was doing well clinically while receiving monthly mineralocorticoids and daily glucocorticoids. This case report is the first to describe the chronological association between alpha-amanitin-induced liver dysfunction and the subsequent development of adrenal insufficiency in a dog.

## 1. Introduction

Toxic mushrooms are found worldwide, and it is estimated that there are over 1000 *Amanita* species that comprise a wide variety of both edible and toxic species [1]. Species can contain amatoxins, phallotoxins, and potentially unidentified compounds with toxic potential [2]. *Amanita phalloides* are endemic to Northern California and have been identified as the cause of morbidity in both human and dog patients [3,4]. They contain cyclic-peptide amatoxins, which result in morbidity and mortality in humans and animals. Following ingestion, the principal lethal toxin found within the genera, alpha-amanitin, inhibits RNA polymerase II and leads to death in rapidly dividing cells, causing the typical clinical presentation of diarrhea, vomiting, and marked hepatic injury [5,6,7]. Given the nonspecific signs of mushroom intoxication and the difficulty differentiating between toxic and non-toxic mushrooms, pet owners may not be proactive about seeking veterinary care for their pets immediately after mushroom ingestion.

In one retrospective study, 13/59 dogs (26%) with confirmed alpha-amanitin exposure survived discharge, and none of these patients were documented to have developed adrenal insufficiency after intoxication at follow-up appointments [3]. Smaller studies have documented longer follow-up periods without evidence of adrenal disease [8,9]. A clearer understanding of post-intoxication outcomes would help veterinarians prepare pet owners to understand the implications and long-term consequences of amanitin intoxication. This case report is the first to describe the chronologic association between alpha-amanitin-induced liver dysfunction and subsequent development of adrenal insufficiency in a dog.

## 2. Case Presentation

A 10-year-old, female spayed Labrador retriever was referred for acute hepatopathy and urinary retention. The dog was reported to have a one-to-two-day history of anorexia, vomiting, and decreased urination with stranguria. There was reported exposure to a variety of unidentified mushrooms in the yard and the dog had a history of dietary indiscretion. However, the patient was not observed to have consumed any mushrooms. Prior to this illness, the patient had an unremarkable medical history with periodic routine laboratory assessment that was within normal limits four months prior.

On presentation (day 0) to the William R. Pritchard Veterinary Medical Teaching Hospital (VMTH), the dog was alert and responsive with heart rate and respiratory rate to be within normal limits. Her abdomen was tense, and a large bladder was palpated. A point-of-care ultrasound revealed a large urinary bladder without evidence of obvious mechanical obstruction. A urinary catheter was placed. A blood gas analysis revealed a mixed acid-base disorder with a normal pH (7.407) characterized by moderate respiratory alkalosis and metabolic acidosis due to mild hyperlactatemia (3.4 mmol/L, RR 0–1.0 mg/dL).

On day 1, a complete blood count revealed a polycythemia with a hematocrit of 62.3% (RR: 40–55%), a moderate thrombocytopenia of 70,000 platelets/μL with clumps (RR: 150,000–400,000/μL), and an inflammatory leukogram characterized by a total white blood count within reference range but the presence of 485 bands with slight toxicity and monocytopenia (121, RR: 150–1200/μL). Biochemistry panel found a severe mixed, primarily hepatocellular hepatopathy with elevated activities of alanine transaminase (ALT) at 13,288 IU/L (RR: 21–72 IU/L), aspartate transaminase (AST) at 1830 IU/L (RR: 20–49 IU/L), alkaline phosphatase (ALP) at 1508 IU/L (RR: 14–91 IU/L), and gamma-glutamyl transferase (GGT) at 17 IU/L (RR: 0–5 IU/L). There was evidence of hepatic insufficiency with a mild hyperbilirubinemia of 1.4 mg/dL (RR: 0–0.2 mg/dL), mild panhypoproteinemia (albumin of 3.3 g/dL (RR: 3.4–4.3 g/dL), globulin of 1.6 g/dL (RR: 1.7–3.1 g/dL)), and mild hypoglycemia of 73 mg/dL (RR: 84–118 mg/dL). Serum cholesterol and blood urea nitrogen were normal. A urine sample collected via cystocentesis was submitted for analysis and culture. Urinalysis revealed a specific gravity of 1.020 with 3 mg/dL bilirubin (RR: 0–2 mg/dL), few transitional epithelial cells, and few squamous epithelial cells. Urine culture revealed a broadly susceptible coagulase-negative *Staphylococcus* (3 × 10^3^ CFU/mL) and non-hemolytic *Escherichia coli* (1 × 10^2^ CFU/mL) that were interpreted to represent contamination from recent urinary catheterization. Urine collected via catheterization was negative for the detection of the B-RAF mutation. This was submitted to evaluate for evidence of a urothelial carcinoma due to her marked urinary bladder distention and reported stranguria. A coagulation panel found a moderately prolonged prothrombin time (PT) of 21.4 s (RR: 7.0–9.3 s) and activated partial thromboplastin time (aPTT) of 21.0 s (RR:10.4–12.9 s) as well as hypofibrinogenemia (<50 mg/dL; RR:109–311 mg/dL) and elevated D-dimers of 904 ng/dL (RR:0–186 ng/dL).

Also on day 1, thoracic radiographs were obtained and interpreted by a board-certified radiologist as unremarkable. An abdominal ultrasound performed by a board-certified radiologist found mildly hyperechoic hepatic parenchyma with few small, hypoechoic nodules, mildly enlarged hepatic and periportal lymph nodes, and changes consistent with gall bladder wall edema suggestive of an acute hepatopathy. The urinary bladder was distended, and the proximal urethra was moderately thickened. The urinary catheter was in place during the ultrasound. The adrenal glands were enlarged bilaterally with the left adrenal gland measuring 1.1 cm in diameter with a slightly heterogeneous architecture, and the right adrenal gland measuring 1.1 cm in diameter with the cranial aspect expanded by an ill-defined rounded hypoechoic region measuring 1.1 cm in diameter (Figure 1).

On day 2, urine obtained on day 0 was submitted for toxicological screening with a previously published liquid chromatography-mass spectrometry assay which has a reportable limit of 3.3 ppb [10]. This assay detected a trace concentration of alpha-amanitin in the specimen, confirming exposure and supporting intoxication.

Given the history and initial diagnostics, amanitin intoxication leading to acute liver failure was prioritized as the top differential. The dog was hospitalized with intravenous fluid support supplemented with dextrose (lactated ringer solution 40–60 mL/h with potassium qs 20 mEq/L, with 2.5% dextrose), methadone for analgesia due to osteoarthritis (0.2 mg/kg IV q 8 h for 8 days), vitamin K supplementation (phytonadione 0.9 mg/kg PO q 12 h), acetylcysteine (70 mg/kg IV q 6 h), antinausea/antiemetic therapy (maropitant 1 mg/kg IV q 24 h and ondansetron 0.5 mg/kg IV q 12 h), and ampicillin/sulbactam (30 mg/kg IV q 8 h) for antimicrobial therapy due to the unknown significance of the bacteriuria. The dog’s urinary catheter was removed on day 5 and she was able to urinate normally with no evidence of urinary retention or lower urinary tract signs.

Serial bloodwork (see Table 1) found decreasing liver enzymes and initially, there was concurrent evidence of progressive hepatic dysfunction. Hepatic function, including coagulation testing, did begin to improve with continued marked improvement of liver enzyme activity.

On the day of discharge (day 8), the dog had a moderate mixed hepatopathy with ALT of 1674 IU/L, AST of 73 IU/L, ALP of 1550 IU/L, and GGT of 7 IU/L, with a hyperbilirubinemia of 0.5 mg/dL, a mild hypoalbuminemia of 3.3 g/dL but normal serum globulins (2.9 g/dL), and a moderate hypocholesterolemia of 104 mg/dL (RR: 139–353 mg/dL). A coagulation panel revealed normalized clotting times with a PT of 8.2 s, aPTT of 13.9 s, fibrinogen of 210 mg/dL, and D-dimer of 181 ng/dL. She was transitioned off dextrose supplementation and remained euglycemic albeit hyporexic at discharge. Her body weight was 27.95 kg. The dog was discharged with amoxicillin/clavulanic acid (13.5 mg/kg PO q 12 h), phytonadione (1.8 mg/kg PO q 12 h), s-adenosylmethionine (SAMe)/silybin (15.3 mg/kg and 1.3 mg/kg, respectively PO q 24 h, Denamarin: Nutramaxx Laboratories, Lancaster, SC, USA), ondansetron (0.3 mg/kg PO q 12 h), ursodiol (9.0 mg/kg PO q 24 h), and gabapentin (7.2 mg/kg PO) as needed for pain.

On day 11, the patient had a decreased appetite and was lethargic. Four days later (day 15) the dog was presented for a recheck appointment. Physical examination documented that she had lost 2.4 kg since discharge. Recheck laboratory assessment revealed improving liver enzymology and improvement in most liver function parameters (Table 2). Her owners were instructed to discontinue phytonadione and SAMe/silybin to improve appetite. However, on day 21, the dog was still hyporexic and lethargic so capromorelin (2.7 mg/kg PO q 24 h, Entyce, Elanco Animal Health, Greenfield, IN, USA) was prescribed.

On day 24, the patient was presented to the VMTH for anorexia and vomiting. On physical examination, the dog was noted to have lost an additional 2.3 kg since her last visit (a total of 4.7 kg since day 0). Her vital signs were within normal limits and initial laboratory assessment revealed a moderate eosinophilia (1806/μL, RR: 0–1500/μL), thrombocytopenia with clumps noted and an otherwise normal leukogram. Chemistry panel showed discordant sodium-to-potassium ratio (25.7) with a hyponatremia of 131 mmol/L (RR: 143–151 mmol/L) and hyperkalemia of 5.1 mmol/L (RR: 3.6–4.8 mmol/L), total hypercalcemia of 11.7 mg/dL (RR: 9.6–11.2 mg/dL), mild hyperphosphatemia of 5.6 mg/dL (RR: 2.6–5.2 mg/dL), severe hypoglycemia of 43 mg/dL, a mild hypoalbuminemia of 3.3 g/dL, a severe hypocholesterolemia of 78 mg/dL, mild hyperbilirubinemia of 0.3 mg/dL, and a mild mixed primarily cholestatic hepatopathy (ALT 85, AST 84, and ALP 447) (Table 3). The patient was admitted and hospitalized for supportive care.

The following day (day 25), the patient was observed to be dull and weak. She was hypoglycemic (69 mg/dL), so a 50% dextrose bolus was administered, and a dextrose continuous rate infusion was initiated. Subsequent recheck blood glucose values were normal. Abdominal ultrasound showed mild submucosal and muscularis thickening of the small bowel with preserved mural architecture and bilateral adrenal nodules (Figure 2), similar in appearance compared to prior. The liver was progressively and diffusely hypoechoic with rounded margins while the cholecystic changes had resolved.

Vitamin B12 and folate were measured and found to be within normal limits. An adrenocorticotropic hormone (ACTH) stimulation was performed using 5 μg/kg cosyntropin administered IV and cortisol results were consistent with adrenal insufficiency (pre: 0.4 μg/dL RR: 0.0–6.0 μg/dL, post: 0.4 μg/dL, RR: 6.0–15.0 μg/dL). The patient was administered dexamethasone-sodium phosphate 0.1 mg/kg IV q12 h and deoxycorticosterone pivalate (DOCP) 2.2 mg/kg IM once. The dexamethasone dose was tapered prior to discharge and the dog was transitioned to oral prednisone. Recheck laboratory assessment showed improvement in the patient’s hyperbilirubinemia, resolved hypoglycemia without dextrose supplementation, and resolved electrolyte derangements (Table 3). The patient had persistent hypoalbuminemia and hypocholesterolemia. By day 27, the patient was bright and alert and eating and drinking reliably on oral medications. She was discharged with instructions to administer oral prednisone 0.4 mg/kg by mouth every 24 h and DOCP injection under the skin every 28 days. The dog was last examined at the teaching hospital 106 days post initial presentation and was doing well clinically, receiving 1.2 mg/kg DOCP injected under the skin monthly and 0.05 mg/kg prednisone by mouth daily.

Given the diagnosis of hypoadrenocorticism during the second hospitalization, a frozen citrated plasma sample stored from day 1 was submitted to the Michigan State Diagnostic Laboratory for analysis of plasma hormone analysis. Plasma aldosterone was reported to be 42 pmol/L (RR: 14–957 pmol/L) and cortisol was reported to be 2.0 μg/dL.

## 3. Discussion

The dog in this case report was diagnosed with amanitin intoxication and subsequent hepatic dysfunction based on the detection of a trace concentration of alpha-amanitin in urine. She recovered and was diagnosed with typical hypoadrenocorticism (HA) 25 days post-initial presentation. The chronology of these events suggests that alpha-amanitin intoxication might have resulted in an adrenocortical injury leading to HA. Given the normal plasma aldosterone concentration on day 1, the patient likely did not have aldosterone deficiency at this time, but this result does not rule out pre-existing cortisol deficiency. Hypoadrenocorticism could have developed in this dog independent of the intoxication caused by idiopathic destruction, immune-mediated or metastatic disease or an ischemic event [11,12,13]. However, the dog’s initial and subsequent diagnostic work-up revealed no evidence of embolic disease or neoplasia, making these differentials unlikely. The initial ultrasound evaluation of the dog revealed mild bilateral adenomegaly, with measurements greater than typically reported for dogs with HA [14,15]. This finding could represent active adrenalitis or may be an incidental finding as the adrenal glands appeared and measured similarly at recheck ultrasound examination on day 25. Without histopathological evaluation of this dog’s adrenal glands, the definitive cause for the development of HA remains a conjecture, but amanitin intoxication is a top differential.

No definitive cause of urinary retention in this patient was identified; however, functional urinary retention due to systemic illness is possible. There was no evidence of urinary retention at subsequent visits beyond the dog’s initial hospitalization.

There are limited published data regarding the development of adrenal insufficiency in other species after intoxication with amatoxin-containing mushrooms. One case series describes adrenal pathology in humans identified during an autopsy after suspected *Amanita phalloides* ingestion. Findings include round cell infiltration of the medulla and vacuolization of the zona fasciculata in one person and congestion of the medulla as well as fatty changes to the cortical cells in the second person [16]. Necropsies of young rats performed 2.5 h post-injection with high concentrations of alpha-amanitin showed significant mitochondrial and nuclear changes to the adrenal fasciculata cells. However, these changes were not observed in rats that were necropsied 24 h post-injection [6].

The evidence that alpha-amanitin can result in adrenocortical injury in humans and rats suggest that it can have similar effects in dogs. In two case reports, the necropsies of dogs who succumbed to suspected *Amanita* intoxication either did not find adrenal gland lesions or did not evaluate the adrenal glands on histopathology [17,18]. One of these dogs was found to have ingested *Amanita muscaria* and developed neurologic signs while the other dog was diagnosed with alpha-amanita intoxication, but the species was not identified. In another study, dogs that survived to discharge after intoxication were reported to have normal biochemical parameters at re-examination with no evidence of HA; however, cortisol and aldosterone were not specifically measured in any dogs [3]. It should be noted that the mortality rate associated with amanitin intoxication in that study was 74% and some dogs that died or were euthanized could have potentially developed HA (or other comorbidities) had they survived. Alternatively, only some mushroom derived toxins may be associated with adrenal injury. The species that caused the intoxication of the dog in our report was not identified.

Hepatoadrenal syndrome may alternatively explain the development of HA in this dog. In humans, hepatoadrenal syndrome has been suggested as a cause of adrenal insufficiency and several studies have shown a high prevalence of this condition in patients with liver disease [19,20,21]. For example, in patients with chronic liver disease, the prevalence of adrenal insufficiency ranged from 29–66%, whereas in acute liver failure, the prevalence ranged from 33–34.6% [19,21,22]. In patients that received a liver transplant without the use of corticosteroid immune suppression, the prevalence ranged from 40–61% [19,21]. A proposed mechanism for hepatoadrenal syndrome is that hepatic dysfunction results in impaired cortisol synthesis due to decreased production of high-density lipoprotein (apoprotein-1). Additionally, the presence of endotoxin and proinflammatory mediators might also inhibit cortisol synthesis [19]. The dog in our report had severe hypocholesterolemia during her initial hospitalization, which worsened at the time of her diagnosis of adrenal insufficiency. Though hepatoadrenal syndrome remains a possibility for the cause of her HA, it is less likely given the rapid improvement and normalization of all biochemical parameters after receiving low-dose (physiologic) glucocorticoid supplementation (Table 3).

## 4. Conclusions

This report highlights important considerations for veterinarians who manage patients that survive amanitin intoxication. Post-recovery from amanitin intoxication, veterinarians should be observant of the development of HA. Since HA can mimic hepatic insufficiency, it is important that this potential consequence of amanitin intoxication be recognized to hasten diagnosis and appropriate therapy for affected dogs. Future areas of investigation include hypothalamic–adrenal axis testing in dogs with hepatic dysfunction to investigate the prevalence of hepatoadrenal syndrome in dogs with suspected amanitin intoxication and other acute hepatic injuries. Additionally, histopathological examination of adrenal glands after amatoxin-related death might help elucidate the link between HA and amanitin intoxication.

## Figures and Tables

**Figure 1 vetsci-10-00500-f001:**
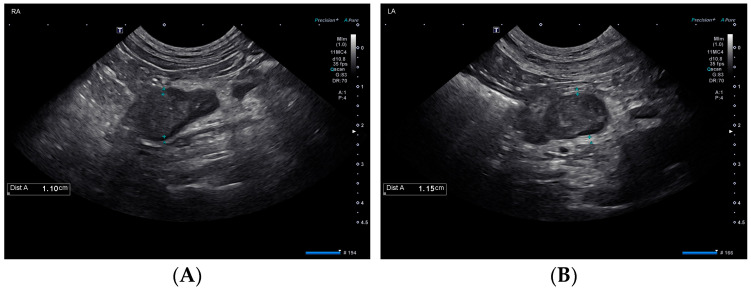
Ultrasonographic findings of the right (**A**) and left (**B**) adrenal glands on day 1.

**Figure 2 vetsci-10-00500-f002:**
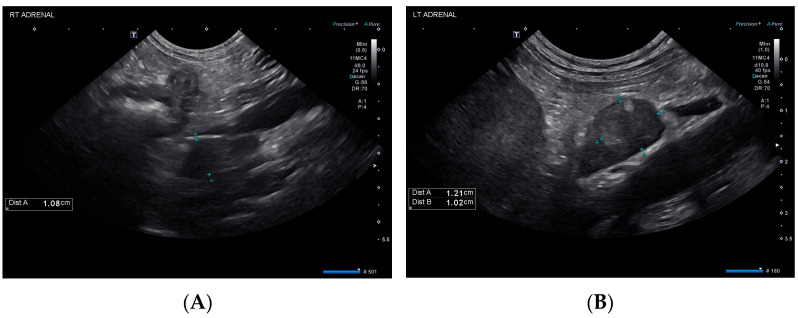
Ultrasonographic findings of the right (**A**) and left (**B**) adrenal glands on day 25.

**Table 1 vetsci-10-00500-t001:** Clinicopathologic parameters throughout initial hospitalization (days 1–8).

	Reference Interval	Day 1	Day 2	Day 3	Day 4	Day 5	Day 6	Day 7	Day 8
Hematocrit (%)	50–55	62.3	60.6				55.3		
Neutrophils (cells/μL)	3000–10,500	8847					8204		
Lymphocytes (cells/μL)	1200–4000	2181					1875		
Eosinophils (cells/μL)	0–1500	485					527		
Platelets (cells/μL)	150,000–400,000	70,000 (clumps seen)					126,000 (clumps seen)		
Sodium (mmol/L)	143–151	148			145	151			
Potassium (mmol/L)	3.6–4.8	4.4			4.5	4.7			
ALT (IU/L)	21–72	13,288	10,343	6007	4103	3270	2555	1993	1674
AST (IU/L)	20–49	1830	467	167	135	110	73	69	73
ALP (IU/L)	14–91	1508	1783	1567	1387	1502	1509	1465	1550
GGT (IU/L)	0–5	17	18	13	8	11	20	5	7
Albumin (g/dL)	3.4–4.3	3.3	3.3	3	2.6	2.8	2.9	3.1	3.3
Globulin (g/dL)	1.7–3.1	1.6	2.1	1.9	1.9	2.2	2.4	2.5	2.9
Cholesterol (mg/dL)	139–353	188	193	137	105	101	99	99	104
Total Bilirubin (mg/dL)	0.0–0.2	1.4	2.1	1.4	0.9	0.6	0.5	0.5	0.5
Glucose (mg/dL)	86–118	73	87	74	149	82	90	63	51
Creatinine (mg/dL)	0.8–1.5	0.8			0.7	0.9			
Blood Urea Nitrogen (mg/dL)	11–33	17	9	5	4	8	10	10	15
PT (s)	7.0–9.3	21.4					9.9		8.2
aPTT (s)	10.4–12.9	21					15.7		13.9
Fibrinogen (mg/dL)	109–311	<50					119		210
D-dimers (ng/dL)	0–186	904					58		181

**Table 2 vetsci-10-00500-t002:** Clinicopathologic parameters on day 15.

	Reference Interval	Day 15
ALT (IU/L)	21–72	214
AST (IU/L)	20–49	33
ALP (IU/L)	14–91	893
GGT (IU/L)	0–5	<3
Albumin (g/dL)	3.4–4.3	3.7
Globulin (g/dL)	1.7–3.1	2.5
Cholesterol (mg/dL)	139–353	100
Total Bilirubin (mg/dL)	0.0–0.2	0.2
Glucose (mg/dL)	86–118	91
Creatinine (mg/dL)	0.8–1.5	
Blood Urea Nitrogen (mg/dL)	11–33	39
PT (s)	7.0–9.3	8.3
aPTT (s)	10.4–12.9	15.2
Fibrinogen (mg/dL)	109–311	160
D-dimers (ng/dL)	0–186	72

**Table 3 vetsci-10-00500-t003:** Clinicopathologic parameters during the second hospitalization (days 24 and 26) and at recheck appointments after discharge (days 34 and 50).

	Reference Interval	Day 24	Day 26	Day 34	Day 50
Hematocrit (%)	50–55	56.9			
Neutrophils (cells/μL)	3000–10,500	6261			
Lymphocytes (cells/μL)	1200–4000	3010			
Eosinophils (cells/μL)	0–1500	1806			
Platelets (cells/μL)	150,000–400,000	34,000 (clumps seen)			
Sodium (mmol/L)	143–151	131	134	154	152
Potassium (mmol/L)	3.6–4.8	5.6	5.5	4.0	3.9
ALT (IU/L)	21–72	85	69	174	108
AST (IU/L)	20–49	84	50	40	26
ALP (IU/L)	14–91	447	356	968	353
GGT (IU/L)	0–5	<3	<3	<3	<3
Albumin (g/dL)	3.4–4.3	3.3	3.1	3.3	3.7
Globulin (g/dL)	1.7–3.1	2.6	2.1	2.0	2.3
Cholesterol (mg/dL)	139–353	78	68	106	164
Total Bilirubin (mg/dL)	0.0–0.2	0.3	0.2	<0.2	<0.2
Glucose (mg/dL)	86–118	43	120	117	114
Creatinine (mg/dL)	0.8–1.5	0.8	0.6	0.6	
Blood Urea Nitrogen (mg/dL)	11–33	28	19	7	8

## Data Availability

The data presented in this case report are available in the article and Appendix A.

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
