# Peer review of "Hypoadrenocorticism in a Dog Following Recovery from Alpha-Amanitin Intoxication"

_vetsci, 2023, doi:10.3390/vetsci10080500_

Round 1

Reviewer 1 Report

Line 58: There was reported exposure to a variety of unidentified mushrooms in the yard: Timing of the exposure? Was the exposure observed?

Line 60: Prior to this illness the patient had an unremarkable medical history with periodic routine laboratory assessment that was within normal limits: When was the dog evaluated last?

Line 63: What does appropriate vital signs mean? Within normal limits? Which vital signs (HR, RR)?

Line 87: Indicate that this analysis was to evaluate if there was evidence of a urothelial carcinoma – it’s not clear why this was suspected based on PE and ultrasound findings.

Line 95: Timing of the ultrasound evaluations and findings is not clear. Once the catheter was places were additional ultrasounds performed and was the bladder distended again?

Line 101: Provide methods of analysis and actual results from toxicologic analysis.

Line 105: Provide dose and duration of methadone treatment – note opiod use in human medicine has been linked with effects on adrenal gland function: Lee AS, Twigg SM. Opioid-induced secondary adrenal insufficiency presenting as hypercalcaemia. Endocrinol Diabetes Metab Case Rep. 2015;2015:150035. doi: 10.1530/EDM-15-0035. Provide drug regimens for all other treatments.

Line 129: Provide BW at discharge

Line 202: Other mushroom species also produce alpha amanitin (e.g., Galerina marginata and Conocybe filaris)

Line 203: Romano et al did not indicate whether adrenals were evaluated.

Line 209: Conjecture about the development of HA in this cohort – they could have also have developed any number of other problems.

Line 225: Though hepatoadrenal syndrome remains a possibility for the cause of her HA… The authors have dismissed this possibility too casually – additional support is needed.

General comment: Bladder distension was not discussed - relevance to possible mushroom poisoning?

Author Response

Response to Reviewers:

Thank you so much for taking the time to review our manuscript. The authors appreciate the thought and effort that went into the review. Below are our changes in response to the comments.

Reviewer 1:

Line 58: There was reported exposure to a variety of unidentified mushrooms in the yard: Timing of the exposure? Was the exposure observed?

Author Response: Thank you for this comment. We have made changes to the paper to include that the patient’s exposure was not observed thus the timing is uncertain.

Line 60: Prior to this illness the patient had an unremarkable medical history with periodic routine laboratory assessment that was within normal limits: When was the dog evaluated last?

Author Response: The paper was updated to reflect that the patient was last evaluated four months prior.

Line 63: What does appropriate vital signs mean? Within normal limits? Which vital signs (HR, RR)?

Author Response: We have clarified to clarify that she had a heart rate and respiratory rate within normal limits.

Line 87: Indicate that this analysis was to evaluate if there was evidence of a urothelial carcinoma – it’s not clear why this was suspected based on PE and ultrasound findings.

Author Response: Thank you for this comment, the authors have clarified that the patient had her urine evaluated due to her history of stranguria and bladder distention.

Line 95: Timing of the ultrasound evaluations and findings is not clear. Once the catheter was places were additional ultrasounds performed and was the bladder distended again?

Author Response:  Thank you for pointing out this area for potential confusion. We have added more detail to clarify the timing of ultrasounds and bladder size. The first ultrasound was a point-of-care ultrasound in the emergency room on day 0, while the second ultrasound was performed by a boarded radiologist on day 1.

Line 101: Provide methods of analysis and actual results from toxicologic analysis.

Author Response:  The assay used is a liquid chromatography-mass spectrometry test that has been previously reported. The actual results of the assay were trace positivity as they fell below the lowest reportable limit.

Line 105: Provide dose and duration of methadone treatment – note opiod use in human medicine has been linked with effects on adrenal gland function: Lee AS, Twigg SM. Opioid-induced secondary adrenal insufficiency presenting as hypercalcaemia. Endocrinol Diabetes Metab Case Rep. 2015;2015:150035. doi: 10.1530/EDM-15-0035. Provide drug regimens for all other treatments.

Author Response: Thank you for this point of clarification. We have included dose/duration/route of treatment for the methadone, as well as included drug regimens for all other treatments.

Line 129: Provide BW at discharge

Author Response: We are uncertain whether you are requesting her body weight or her blood work. The dog weighed 27.95kg at discharge. This has been added to the text. Additionally, labs have been moved from supplemental data to the manuscript per request of the editors.

Line 202: Other mushroom species also produce alpha amanitin (e.g., Galerina marginata and Conocybe filaris)

Author Response: Thank you for drawing attention to this. We have changed the wording to indicate it was suspected Amanitaintoxication.

Line 203: Romano et al did not indicate whether adrenals were evaluated.

Author Response: Thank you for this comment. We have updated the paper to reflect that the adrenal glands may have not been evaluated.

Line 209: Conjecture about the development of HA in this cohort – they could have also have developed any number of other problems.

Author Response: Thank you for bringing up this point. The authors have included changes to indicate that the patients who were euthanized or died could have developed HA or other comorbidities had they survived.

Line 225: Though hepatoadrenal syndrome remains a possibility for the cause of her HA… The authors have dismissed this possibility too casually – additional support is needed.

Author Response: Hepatoadrenal syndrome is a poorly described condition in humans and has not been reported in dogs. However, in the cases reported and summarized in the most comprehensive systematic review (Adrenocortical dysfunction in liver disease: a systematic review. Fede et al. Hepatology 2012 Vol. 55 Issue 4 Pages 1282-91) the adrenal insufficiency resulting from various hepatic insults is strictly glucocorticoid deficiency. This syndrome could manifest differently in dog, but the mineralocorticoid deficiency in this case sets it apart from what has been identified in humans. In dogs, reports of relative adrenal insufficiency do exist (Relative adrenal insufficiency in dogs with sepsis. Burkitt, et al. J Vet Intern Med 2007 Vol. 21 Issue 2 Pages 226-3), but again these reports note only glucocorticoid deficiency.

General comment: Bladder distension was not discussed - relevance to possible mushroom poisoning?

Author Response: Thank you for bringing this point to our attention. We have added a statement to the discussion regarding the bladder distention.

Reviewer 2 Report

In this manuscript, Cohen et al report the development of HA few days after alpha-amanitin intoxication in a female Labrador retriever. This is a very interesting and useful study, as alpha-amanitin intoxication could be included in the differential diagnosis of HA and vice versa, dogs that are presented with amanitin intoxication should be monitored for the potential development of HA. I will kindly ask the authors to specify that in line 158, the values in the parentheses refer to cortisol levels.

Author Response

Response to Reviewers:

Thank you so much for taking the time to review our manuscript. The authors appreciate the thought and effort that went into the review. Below are our changes in response to the comments.

Reviewer 2:

I will kindly ask the authors to specify that in line 158, the values in the parentheses refer to cortisol levels.

Author Response: Thank you for this comment. We have clarified that these values were cortisol levels.

Round 2

Reviewer 1 Report

Concerns have been addressed.

Quality of English language is good - no concerns.